# Modelling the Molecular Permeation through Mixed-Matrix Membranes Incorporating Tubular Fillers

**DOI:** 10.3390/membranes11010058

**Published:** 2021-01-14

**Authors:** Ali Zamani, F. Handan Tezel, Jules Thibault

**Affiliations:** Department of Chemical and Biological Engineering, University of Ottawa, Ottawa, ON K1N 6N5, Canada; azama049@uottawa.ca (A.Z.); Handan.Tezel@uottawa.ca (F.H.T.)

**Keywords:** mixed-matrix membranes (MMMs), butanol separation, tubular fillers, nanotubes, finite differences, polydimethylsiloxane membrane, membrane permeability

## Abstract

Membrane-based processes are considered a promising separation method for many chemical and environmental applications such as pervaporation and gas separation. Numerous polymeric membranes have been used for these processes due to their good transport properties, ease of fabrication, and relatively low fabrication cost per unit membrane area. However, these types of membranes are suffering from the trade-off between permeability and selectivity. Mixed-matrix membranes, comprising a filler phase embedded into a polymer matrix, have emerged in an attempt to partly overcome some of the limitations of conventional polymer and inorganic membranes. Among them, membranes incorporating tubular fillers are new nanomaterials having the potential to transcend Robeson’s upper bound. Aligning nanotubes in the host polymer matrix in the permeation direction could lead to a significant improvement in membrane permeability. However, although much effort has been devoted to experimentally evaluating nanotube mixed-matrix membranes, their modelling is mostly based on early theories for mass transport in composite membranes. In this study, the effective permeability of mixed-matrix membranes with tubular fillers was estimated from the steady-state concentration profile within the membrane, calculated by solving the Fick diffusion equation numerically. Using this approach, the effects of various structural parameters, including the tubular filler volume fraction, orientation, length-to-diameter aspect ratio, and permeability ratio were assessed. Enhanced relative permeability was obtained with vertically aligned nanotubes. The relative permeability increased with the filler-polymer permeability ratio, filler volume fraction, and the length-to-diameter aspect ratio. For water-butanol separation, mixed-matrix membranes using polydimethylsiloxane with nanotubes did not lead to performance enhancement in terms of permeability and selectivity. The results were then compared with analytical prediction models such as the Maxwell, Hamilton-Crosser and Kang-Jones-Nair (KJN) models. Overall, this work presents a useful tool for understanding and designing mixed-matrix membranes with tubular fillers.

## 1. Introduction

The use of membranes in separation processes for chemical, petrochemical, and environmental applications has significantly increased in recent years. The main advantages of using membranes in industrial separation processes are the much lower energy requirements and the smaller plant footprint compared to conventional separation processes. In addition, given their high stability, efficiency, and ease of processing, employing membranes in industrial processes leads to lower environmental impact and cost [1,2]. Polymeric membranes are currently used commercially in separation processes such as pervaporation [3,4] and gas separation [5]. In gas separation processes, cellulose acetate membranes were employed by Cynara, Separex, and GMS to remove carbon dioxide from natural gas by the mid-1980s and further developed by involving polyimide hollow-fiber membranes [6]. At about the same time, GENERON introduced the first membrane system to separate nitrogen from air using poly(4-methyl-1-pentene) (TPX) membranes. Due to the collaboration of other companies such as Dupont and Air Liquide, over 10,000 similar systems have already been installed worldwide. However, many gas processes such as oxygen separation or pervaporation applications such as alcohol separation and dehydration still require better membranes to become more commercially viable [2,6].

Improving the separation performances of a membrane-based process at both laboratory and industrial scales depends largely on the chemical, mechanical, and permeation properties of the membrane. Even though polymeric membrane materials are continuously improved [7], polymeric membranes mostly suffer from the well-known trade-off between the separation factor and permeating flux [4,8,9]. In an attempt to improve the separation factor and permeating flux of polymeric membranes, many researchers are now directing their efforts to developing mixed-matrix membranes (MMMs). It has been reported that MMMs, which are made by embedding a proper organic or inorganic filler in the polymer matrix, can combine the advantages of the higher selectivity of the filler particles and the ease of processing of polymers [3,4,10,11,12]. Different filler materials, such as activated carbons [4], carbon nanotubes (CNTs) [13], zeolites [14], and metal-organic frameworks (MOFs) [15,16], have been incorporated within the matrix of polymers to make MMMs.

Among all types of fillers, nanotubes are considered emerging nanostructured materials for their potential to enhance the separation performance of membranes for numerous applications. Since the discovery of carbon nanotubes (CNTs), mixed-matrix membranes incorporating different single-wall and multi-wall carbon nanotubes [17,18] have shown their potential due to their exceptional transport properties and their physical compatibility with the polymeric membrane. Although the mass transport properties of CNTs are appealing, the ability to mass-produce and fabricate defect-free mixed-matrix membranes using CNTs is still challenging and limits their applications to large-scale industrial processes. To address these problems, other nanotubes such as titania [19], halloysite nanotubes [20,21], organosilicon [22], and aluminosilicate [23,24] nanotubes have been investigated. Novel techniques have been developed for the synthesis of nanotubes as well as to manipulate their dimension [25,26] and to modify their surface functionality [27,28]. Recent numerical studies on nanotubes have suggested that they possess mass transport rates up to three orders of magnitude larger than other materials with similar channel sizes, such as zeolites. The mass transport rates were also found to be considerably larger than the one predicted based on the Knudsen diffusion [29,30]. However, most nanotubes have an impermeable side wall that causes the orientation of nanotubes to greatly impact the achievable mass transport, as the permeant can only diffuse in the nanotube axial direction [31,32,33]. Since Skoulidas et al.’s [34] atomistic simulations on vertically aligned CNTs demonstrated the extremely high transport rate and permeability of light gases, aligning nanotubes within the membrane has received considerable attention for the development of membrane-based separation technologies.

Although there are numerous types of nanotubes and polymers available, a rational choice of both phases toward the preparation of MMMs is necessary. Therefore, theoretical predictions of the separation performance from the pure species permeation properties in these MMMs become increasingly valuable. Up to now, different theoretical models have been developed to predict the performance of both ideal and non-ideal MMMs based on their polymer-particle interface morphology [35,36]. Different models, including the modified Maxwell model proposed by Vu et al. [37], the modified Lewis−Nielsen model [2,36], the modified Pal model [36], as well as the original and modified Felske model [35], have been developed to estimate the effective permeability of non-ideal MMMs. Generally, these models can predict the permeability and the selectivity for the most common MMM morphologies over a large range of filler loading [2]. However, several additional parameters such as particle pore size and distribution, permeability of species in the rigidified or void layer, filler-pore blockage ratio, as well as the thickness of the non-ideal phase should be taken into account. These parameters are most sensitive to operating conditions and, since there are no reliable methods yet to measure these parameters, their estimation and determination remain a significant challenge [1]. Concentration gradient-driven molecular dynamics simulations, although currently limited to short simulation times, can contribute to elucidating the interaction between the polymer and the filler and estimate transport properties near and at the filler-polymer interphase region [38].

Even though the ideal polymer-particle interface is generally difficult to achieve, the non-ideality can be negligible in some MMMs depending on the particle size and intrinsic properties of polymer and particle materials [1,2]. Numerous analytical models have been developed for estimating the effective permeability of ideal MMMs with spherical or near-spherical fillers such as activated carbons, zeolites, or metal-organic frameworks (MOFs) [1,2,8,39]. The Maxwell model [40] is a well-known correlation to predict effective permeability in terms of the permeability of the dispersed and continuous phases and the filler volume fraction:(1)$$\frac{P_{eff}}{P_{c}}=\frac{P_{d}+(n-1)P_{c}+(n-1)(P_{d}-P_{c})\phi }{P_{d}+(n-1)P_{c}+(P_{d}-P_{c})\phi }$$
where *P_eff_* is the effective permeability of the mixed-matrix membrane, *P_c_* is the permeability of the polymer matrix (continuous phase), *P_d_* is the permeability of the filler (dispersed phase), *φ* is the volume fraction of the filler 0 ≤ *φ* ≤ 1, and *n* is the shape factor of the filler. Considering the original Maxwell model for near-spherical fillers, the shape factor *n* is taken as *n* = 3.

The Maxwell correlation can predict the effective permeability of MMMs for spherical and near-spherical fillers fairly accurately up to the intermediate filler volume fraction [1]. However, the Maxwell correlation is not well adapted for predicting the effective permeability of particles that deviate significantly from traditional geometry such as for MMMs incorporating tubular fillers, as it is difficult to specify the shape factor. A modified Maxwell model, also known as the Hamilton-Crosser model [41], was suggested for the prediction of effective permeability by substituting the semi-empirical shape factor in the Maxwell equation using Equations (2) and (3).
(2)$$n=3\psi^{-g}$$
(3)ψ=2(34)23doL2do+2L
where *ψ* represents the sphericity of nanotubes and *g* is an empirical parameter taken as unity, as in the original paper by Hamilton and Crosser [41]. *L* and *d_o_* are the length and outer diameter of the nanotubes, respectively. Although the Hamilton-Crosser model can estimate permeation in composite membranes containing nanotubes by using a shape factor, the assumption that the fillers have an isotropic permeability makes it inconsistent with the high mass transport in the axial direction of the nanotubes. Furthermore, the Hamilton-Crosser model assumes that the fillers are randomly oriented, whereas nanotubes could be well aligned in the mixed-matrix membranes [13].

The Kang-Jones-Nair (KJN) model [30] is among the few analytical models that were proposed to estimate effective permeability considering the orientation of the nanotubes and their permeability in the axial direction (Equations (4)–(6)).
(4)$$\frac{P_{eff}}{P_{c}}={\left[\left(1-\frac{\cos(\theta )}{\cos(\theta )+\frac{1}{\alpha }\sin(\theta )}\phi \right)+\frac{P_{c}}{P_{d}}\left(\frac{1}{\cos(\theta )+\frac{1}{\alpha }\sin(\theta )}\right)\phi \right]}^{-1}$$
(5)$$\phi =\frac{V_{d}}{V_{t}}$$
(6)$$\alpha =\frac{L}{d_{o}}$$
where *V_t_* is the total volume, *V_d_* is the dispersed phase volume based on the length and outside diameter of the nanotube, and *θ* is the orientation with respect to the axis parallel to the main migration direction and nanotube aspect ratio *α*, which is the length over the outer diameter ratio (*L*/*d_o_*).

Even though the KJN model can account for the orientation and aspect ratio of the nanotubes, a one-dimensional (1D) mass transfer was assumed for its derivation. This approximation may not represent adequately the three-dimensional (3D) mass transport phenomena occurring in MMMs and consequently reduces the reliability of the KJN model. In addition, the KJN model cannot be used for completely or partially impermeable filler particles.

To assess the degree of accuracy of the previously mentioned models for the estimation of effective permeability, the three-dimensional Fick’s diffusion equation was solved numerically to determine the steady-state permeation flux of a mixed-matrix membrane containing nanotubes. The steady-state permeation flux allows for the calculation of a given MMM’s effective permeability. The numerical investigation allows the effects of the filler orientation, the length-to-diameter aspect ratio, and the permeability ratio between the continuous and dispersed phases on the effective permeability to be studied. The results were compared with the Maxwell, Hamilton-Crosser, and KJN models and provided data to potentially develop a better empirical predictive model. There were numerous studies in the literature using different types of fillers, where the effective permeability did not follow the expected trends, and many reasons were provided to explain these discrepancies. The relative permeability of ideal MMMs, calculated numerically as in this investigation and used as a benchmark, would be very helpful in identifying the sources of these non-idealities. The objective of this paper is therefore to investigate the impact of using nanotubes as fillers in MMMs on the permeability of a migrating species under ideal filler distribution. Since nanotubes such as CNTs are increasingly used in MMMs in various separation processes, including liquid and gas separation and pervaporation, it is very important to understand fundamentally how the permeability changes with different operating parameters. The data obtained in this investigation can also serve to develop an empirical model for MMM permeability.

## 2. Materials and Methods

### 2.1. Mixed-Matrix Membrane Model Construction

Assuming a homogenous dispersion of the filler particles with the same orientation within the polymeric membrane, as shown in Figure 1a, a mixed-matrix membrane can be described as the repetition of a unit element. Each unit element has an identical effective permeability, which gives an accurate representation of the permeability of the entire mixed-matrix membrane. Hence, it is possible to solve numerically for a unique unit element, as shown in Figure 1, to determine the effective permeability of the MMM. Figure 1b shows that each repeatable unit of the mixed-matrix membrane, which is referred to as a “unit element” in this work, has three distinct regions: (1) the polymeric continuous phase surrounding the filler, (2) the hollow cylindrical filler particle located in the center of the unit, and (3) the cylindrical void phase inside the filler.

### 2.2. Modelling Mass Transfer

The three-dimensional Fick’s second law of diffusion (Equation (7)) was solved numerically for all phases to determine the concentration profile within MMMs as a function of time and under steady state:(7)$$\frac{\partial C}{\partial t}=\frac{\partial }{\partial x}\left(D_{x,y,z}\frac{\partial C}{\partial x}\right)+\frac{\partial }{\partial y}\left(D_{x,y,z}\frac{\partial C}{\partial y}\right)+\frac{\partial }{\partial z}\left(D_{x,y,z}\frac{\partial C}{\partial z}\right)$$
where *C* is the concentration of the migrating species inside the MMM, *D* is the diffusion coefficient in *x*, *y* and *z* directions, and *t* is the time.

In this investigation, it was assumed that the particle-filler interface morphology was ideal. Ideal membrane morphology refers to a two-phase membrane system with good adherence between the continuous phase and the dispersed phase [42], uniform filler dispersion, no interface voids, and no polymer penetration in the tubular section. It is further assumed that the concentration of the feed solution in contact with the retentate side of the membrane remains constant while the perfect vacuum prevails on the permeate side. Constant solubility (*S*) of the migrating species and instantaneous equilibrium are assumed at the fluid-membrane interface. Since it is assumed that the membrane is made of a large number of identical unit elements, periodic boundary conditions (PDC) exist at the four faces parallel to the main permeation direction. Considering that it is the steady-state permeation flux that is of interest in order to calculate effective permeability, a linear concentration profile in the main permeation direction was used as the initial condition for the concentration profile in order to reduce the convergence time required to achieve steady state. It is important to note that the final steady-state concentration profile and permeate flux are independent of the initial conditions. The initial and boundary conditions for the mass transfer simulations used in this investigation are summarized in Table 1.

### 2.3. Numerical Methods and Solution Post-Processing

The three-dimensional dynamic and steady-state concentration distributions within one repeatable unit element of an MMM were obtained by solving numerically Fick’s diffusion equation (Equation (7)) using the finite differences method (FD). The computer code to solve Fick’s diffusion equation was written in FORTRAN. The unit element consisted of a finite hollow cylindrical filler located at the center of a rectangular cuboid. The unit element was discretized into a sufficiently large number of mesh points by implementing OpenFOAM software to generate small cubical elements, each consisting of only one phase (continuous or dispersed phase). The dimensions of the unit element and the filler as well as the discretization mesh size were kept constant. In some cases, the number of filler particles embedded in the unit element was varied from 0 to 4 in order to calculate the concentration profile for different volume fractions of the dispersed phase. For all numerical simulations, the largest possible mesh size was selected, for which the mesh independency prevailed in order to obtain at the same time an accurate solution and lower computation time. Mesh independency implies that the obtained solution was independent of the mesh resolution.

Assuming that the solubility and mass diffusivity of each permeating species remained constant throughout the unit element and the membrane, Equation (8) was used to calculate the concentration of each species (*m*) at mesh point (*i*,*j*,*k*) at the next time step *t* + Δ*t*.
(8)$$C_{i,j,k}^{m,t+\Delta t}=C_{i,j,k}^{m,t}+\Delta t\left[\begin{array}{c} -D_{i,j,k}^{LX}\frac{\left(C_{i,j,k}^{m,t}-\frac{S_{i,j,k}}{S_{i-1,j,k}}C_{i-1,j,k}^{m,t}\right)}{\Delta x^{2}}+D_{i,j,k}^{RX}\frac{\left(\frac{S_{i,j,k}}{S_{i+1,j,k}}C_{i+1,j,k}^{m,t}-C_{i,j,k}^{m,t}\right)}{\Delta x^{2}}\\ -D_{i,j,k}^{LY}\frac{\left(C_{i,j,k}^{m,t}-\frac{S_{i,j,k}}{S_{i,j-1,k}}C_{i,j-1,k}^{m,t}\right)}{\Delta y^{2}}+D_{i,j,k}^{RY}\frac{\left(\frac{S_{i,j,k}}{S_{i,j+1,k}}C_{i,j+1,k}^{m,t}-C_{i,j,k}^{m,t}\right)}{\Delta y^{2}}\\ -D_{i,j,k}^{LZ}\frac{\left(C_{i,j,k}^{m,t}-\frac{S_{i,j,k}}{S_{i,j,k-1}}C_{i,j,k-1}^{m,t}\right)}{\Delta z^{2}}+D_{i,j,k}^{RZ}\frac{\left(\frac{S_{i,j,k}}{S_{i,j,k+1}}C_{i,j,k+1}^{m,t}-C_{i,j,k}^{m,t}\right)}{\Delta z^{2}} \end{array}\right]$$

The concentration change at mesh point (*i*,*j*,*k*) during the time interval Δ*t* only depends on the current concentration at the mesh point and the six neighbouring mesh points at time *t*. Since the dispersed and continuous phases have different properties, the solubility of each phase is different. Therefore, the solubility ratio of each neighbouring mesh point is used to convert all the concentrations to the same phase basis as the phase of the center mesh point (*i*,*j*,*k*). A mass balance in the *x*, *y*, and *z* directions was performed to calculate the effective diffusivity coefficient of each mesh point within the matrix of the membrane due to their solubility and diffusivity differences. Equations (9) and (10) give the equations that were used to calculate the effective diffusivity in the *x*-direction between mesh point (*i*,*j*,*k*) and its left (L) neighbour (*i* − 1,*j*,*k*) and between mesh point (*i*,*j*,*k*) and its right (R) neighbour (*i* + 1,*j*,*k*), respectively. Similar equations were derived for the effective diffusion coefficients in the other two dimensions and were substituted in Equation (8).
(9)$$\frac{1}{D_{i,j,k}^{LX}}=\frac{S_{i,j,k}}{S_{i-1,j,k}}\frac{1}{2D_{i-1,j,k}}+\frac{1}{2D_{i,j,k}}$$
(10)$$\frac{1}{D_{i,j,k}^{RX}}=\frac{S_{i,j,k}}{S_{i+1,j,k}}\frac{1}{2D_{i+1,j,k}}+\frac{1}{2D_{i,j,k}}$$

Fick’s first law of diffusion, given in Equation (11), was used to estimate the average permeation flux of a permeant at the permeate side of the membrane based on all surface mesh points of the *x–z* plane (Equation (12)). A similar equation was used to estimate the permeation flux at the membrane interface on the feed side of the membrane.
(11)$$J_{\left(i,N_{y},k\right)}=-D_{x,y,z}{\frac{\partial C}{\partial y}|}_{y=L_{2}}$$
(12)$${J_{m}|}_{y=L_{2}}=\sum_{i=1}^{N_{x}}\sum_{k=1}^{N_{z}}\frac{\omega_{i}\omega_{k}J_{\left(i,N_{y},k\right)}}{\left(N_{x}-1\right)\left(N_{z}-1\right)}\;\mathrm{with}\;|\begin{array}{c} \omega_{i}=\{\begin{array}{c} 0.5,i=1\;\mathrm{or}\;N_{x}\\ 1,i\in \left[2,N_{x}-1\right] \end{array}\\ \omega_{k}=\{\begin{array}{c} 0.5,k=1\;\mathrm{or}\;N_{z}\\ 1,k\in \left[2,N_{z}-1\right] \end{array} \end{array}$$
where *J_m_* is the average permeation flux of component *m* calculated for a *x*–*z* plane. *N_x_*, *N_y_*, and *N_z_* are the number of mesh points used to discretize Equation (7) in the *x*, *y*, and *z* directions, respectively. Given the estimation of the permeation flux, the concentration difference across the unit element, and the thickness of the unit element, the effective permeability of a permeant in an MMM can be calculated using Equation (13).
(13)$$P_{m,eff}=\frac{J_{m}L_{2}}{C_{m,f}-C_{m,p}}$$
where *L*_2_ is the thickness of the unit element and *C_m,f_* and *C_m,p_* are the concentrations of component *m* at membrane surface on the feed side and permeate side, respectively.

It is important to note that the simulations were run until the difference between the permeation fluxes at the feed and the permeate side of the membrane achieved a relative error smaller than 0.01% to ensure steady-state conditions prevailed. As illustrated in Figure 2, the tubular filler was described by an inner diameter (*d_i_*), a length-to-outside diameter aspect ratio (α) calculated using Equation (6), and orientation defined by an angle *θ* (varying from 0 to 90°) with respect to the axis parallel to the main migration direction (*y*-direction). The diffusion coefficient and solubility factor for the continuous polymer phase were varied for different case studies, whereas the permeation mechanism inside the nanotube was assumed to follow Knudsen diffusion due to the small internal diameter of the tubular nanofillers and the low partial pressure of the migrating species. As a result, the permeability of the nanotube channel (dispersed phase) was estimated using Equation (14).
(14)$$P_{d}=S_{d}D_{d}=\frac{1}{RT}\sqrt{\frac{8d_{i}^{2}RT}{9\pi M}}$$
where *D_d_* is the Knudsen diffusivity, *S_d_* is the solubility factor, *d_i_* is the inner diameter of the nanotube, and *M* is the molecular weight of the transported molecule.

## 3. Results and Discussion

### 3.1. Unit Element Validation

In this section, four different numerical simulation studies were performed to show unequivocally that the effective permeability of a repeatable unit of an MMM is identical to the effective permeability of the whole mixed-matrix membrane with similarly aligned and uniformly distributed nanotubes. In Case I (Figure 3b), a slice of the membrane including four identical unit elements with a nanotube located at their respective center was simulated and the effective permeability of the four unit elements was calculated at steady state. The four unit elements of Case I were then divided into two slices comprised of two unit elements, a horizontal and a vertical slice represented as Cases II (Figure 3c) and III (Figure 3d), respectively. Finally, Case IV (Figure 3e) considered the typical unit element containing a single filler particle used throughout this investigation. For all case studies, simulations were performed for an aspect ratio of α = 10 and a ratio of the permeability of the dispersed phase over the permeability of the continuous phase of *P_d_*/*P_c_* = 100 with all tubular fillers oriented along the *y*-axis and with no species diffusion through the outside wall of the nanotubes. The volume fraction for each case was either 0.1 or 0.4 and was calculated using Equation (5). To keep the generality of the results, the effective permeability was reported for each case study as the relative permeability (*P_r_*) as calculated using Equation (15).
(15)$$P_{r}=\frac{P_{eff}}{P_{c}}$$
where *P_eff_* and *P_c_* represent the effective permeability of the MMM and the permeability of the continuous phase, respectively. All specifications of the dispersed and continuous phases are summarized in Table 2.

Considering the results obtained from the eight numerical simulations of Figure 3b–e and Table 2, the different cases with different dimensions and spatial distribution yielded identical relative permeability. Furthermore, the same results were obtained by performing eight additional simulations with a volume fraction of 0.2 and 0.3. Therefore, each case led to an identical and accurate estimation of the permeability of the entire mixed-matrix membrane. However, as shown in Figure 4, a significant increase in computation time was observed when increasing the simulation dimensions for Cases I, II, and III. Thus, Case IV was chosen to be pursued for the rest of the case studies in this investigation.

### 3.2. Effect of the Filler Orientation on the Effective Permeability

Recent research on the use of nanotubes as a filler in MMMs generally focused on incorporating different types of nanotubes in different polymer matrices in order to achieve a higher effective permeability. Usually, these mixed-matrix membranes are fabricated by incorporating nanotubes without applying an external driving force to align the nanotubes with a preferred orientation. However, the possibility of controlling the filler orientation in mixed-matrix membranes is what makes the nanotube fillers more advantageous over other types of nano-porous materials. The orientation of the fillers is also of paramount importance for other types of fillers such as graphene nanosheets [43] and impermeable cuboids [44].

As clearly shown by Ebneyamini et al. [8], the effective permeability of the membrane for a permeant depends on the permeability of each phase and not on the individual values of the solubility factor and diffusion coefficient but rather on the product of the two. Therefore, to investigate the effect of the orientation of the nanotubes, five numerical simulations of a unit element incorporating a single nanotube with different filler orientations were performed. The five angles tested were 0, 30, 45, 60, and 90°. The permeability ratio of the dispersed phase and the continuous phase was kept constant at a value of *P_d_*/*P_c_* = 100, and the length-to-diameter ratio *α* was set to 10. The results of the effect of the nanotube orientation in terms of the relative permeability (*P_eff_*/*P_c_*) are presented in Figure 5 for three different filler volume fractions. As expected, the results of Figure 5 show that a higher filler volume fraction had a larger effect on the relative permeability. The vertical orientation of high permeability nanotubes provided a longer projected preferential permeation pathway for the molecules to diffuse across the membrane, thereby enhancing the relative permeability of the membrane. Even though the vertical filler provided a smaller surface of higher permeability relative to the impermeable area between the inner and the outer diameters, the effective permeability of the membrane was significantly increased due to the high permeability in the axial direction of the filler. On the other hand, the horizontal orientation presented the lowest effective permeability. As previously mentioned, there is no mass diffusion occurring through the side wall of the nanotubes, so molecular species must contour these impermeable walls to migrate through the membrane. Consequently, the side wall acts as an additional mass transfer resistance that increases by increasing the filler angle from 0° to 90°. The maximum resistance occurs at a horizontal orientation where these fillers act as barrier material. The results in Figure 5 show that the enhancement and reduction in the effective permeability were magnified by increasing the filler volume fraction. For this particular case study, at an angle in the vicinity of 45°, the results show that the enhancement and the reduction in the effective permeability canceled each other out. It is therefore obviously desirable to align the nanotubes in the direction of the permeation of the molecular species if it is desired to increase the membrane permeability.

### 3.3. Effect of the Permeability Ratio of the Nanotubes on the Effective Permeability

To assess the effect of the dispersed-to-continuous phase permeability ratio (*P_d_*/*P_c_*) on the relative permeability *P_r_* of mixed-matrix membranes incorporating nanotubes, a series of numerical simulations was performed for five different nanotube orientations and three filler volume fractions. For all simulations, the length-to-diameter aspect ratio α of the nanotube and the outside-to-inside diameter ratio of the diameter *d_o_*/*d_i_* were 10 and 2.5, respectively. The results for five different permeability ratios (*P_d_/P_c_*) from 0 to 1000 are presented in Figure 6.

The results of Figure 6 show that the relative permeability *P_r_* of the mixed-matrix membrane increased with an increase in the dispersed-to-continuous phase permeability ratio. As observed in the previous section, this permeability improvement is highly dependent on the nanotube orientation and a higher relative permeability is obtained with nanotubes oriented in the direction of species permeation (θ = 0°). For a permeability ratio (*P_d_/P_c_*) of 1000, the relative permeability increased by 43% for vertically aligned nanotubes and a volume fraction of 0.10. The permeability along the axial direction of the nanotubes provides a preferential diffusion pathway for the permeating species to migrate more rapidly. Further increasing the permeability ratio does not lead to a higher relative permeability because the species still have to migrate through the less permeable polymer section before reaching the nanotubes and the polymer matrix becomes the limiting factor in the effective permeability.

The results of Figure 6 highlight a very interesting phenomenon regarding the range of angles of nanotubes for which the relative permeability *P_r_* was above unity, which was completely independent of the volume fraction, and the intersection point with the relative permeability of unity could be referred to as the break-even point. These results show that nanotubes enhanced the effective permeability of the membrane when the orientation of the nanotubes was above the break-even point. For a very high permeability ratio (*P_d_*/*P_c_*), there existed an improvement in the effective permeability of the mixed-matrix membrane for angles up to roughly 50°, and the angle for the break-even point decreased as the permeability ratio decreased. The angle was only 10^o^ when the permeability ratio (*P_d_*/*P_c_*) was equal to 10. The enhancement completely disappeared at a permeability ratio in the vicinity of 5. Above the angle for the break-even point, the nanotubes acted as a barrier material due to the impermeability of the nanotube wall. For a permeability ratio (*P_d_*/*P_c_*) of unity and vertically aligned nanotubes, the relative permeability was below one because of the volume fraction occupied by the impermeable walls of the nanotubes, which offered an additional resistance to mass transfer.

At *P_d_*/*P_c_* = 0, the filler was assumed to be a non-porous solid cylinder. Figure 6 acknowledges the fact that there was no significant difference in the effective permeability between *P_d_*/*P_c_* = 1 and *P_d_*/*P_c_* = 0. The relative permeability $P_{r}$ in both cases was dependent to the orientation *θ* and the filler volume fraction, but it was always less than unity. Therefore, nanotube fillers with lower permeability ratio are not suitable if highly permeable membranes are desired.

To more clearly visualize the break-even point, the MMM relative permeability *P_r_* was plotted as a function of the orientation of the nanotubes for the five permeability ratios (*P_d_*/*P_c_*) and for a constant filler volume fraction of 0.10 (Figure 7). It is clear that for a permeability ratio of 10 or lower, the use of nanotube fillers, even if they were perfectly aligned vertically, did not enhance the effective permeability of the membrane. One could argue that for a permeability ratio of unity, the relative permeability should also be unity. However, the volume occupied by the impermeable walls of the nanotubes acts as a barrier material and the relative permeability will always be below unity. At an angle *θ* of 90° (horizontal orientation), the relative permeability of the MMM becomes independent of the dispersed-to-continuous phase permeability ratio (*P_d_*/*P_c_*), as the nanotubes act strictly as barrier materials and there is no diffusion along the axial direction of the nanotubes.

### 3.4. Effect of the Filler Aspect Ratio of Nanotubes on the Relative Permeability

To investigate the impact of the length-to-diameter aspect ratio (*α*) of the nanotubes on the relative permeability of MMMs, a series of numerical simulations was performed for five aspect ratios varying from 0 to 100 and various filler volume fractions while the permeability ratio (*P_d_*/*P_c_*) and the orientation were kept constant at 100 and 0°, respectively. Figure 8 presents the plots of the relative permeability of MMMs as a function of the filler volume fraction. For each simulation, the diameter of the nanotubes was kept constant and the filler length was varied. These results clearly show that, with the same filler volume fraction, the effective permeability of MMMs was higher for higher aspect ratios. Similar to the results of previous sections, as expected, increasing the filler volume fraction enhanced the effective permeability for all aspect ratios. These results indicated that tubular fillers with higher aspect ratios and vertically aligned are preferred for enhancing the effective permeability of MMMs.

### 3.5. Separation Properties Prediction of Binary Mixtures

The ultimate objective of using mixed-matrix membranes is to enhance the separation factor of gas or liquid mixtures. In this investigation, it was desired to examine how a mixed-matrix membrane with nanotube fillers could be used for the binary mixture separation of butanol and water. Separation of alcohol/water mixtures using pervaporation is one of the most popular applications in this field. In this section, a butanol/water binary mixture was chosen to investigate the effect of the nanotube orientation on the effective permeability and the separation performance for both species. Since the mass diffusivity and solubility coefficient of both components in the polymeric membrane are required, the experimental data of a commercial polydimethylsiloxane (PDMS) membrane was considered. The simulation parameters and details for this analysis are summarized in Table 3. In this investigation, it was assumed that given that the concentrations within the membrane were relatively low, there were no interaction effects between the two migrating species [1]. Since the Knudsen diffusion mechanism was assumed to prevail within the nanotubes, water molecules with a smaller molecular weight should have diffused faster through the filler. Therefore, by rotating the nanotubes from a horizontal to a vertical orientation, the effective permeability of both butanol and water would increase. This increase in the effective permeability should have enhanced the total flux of the membrane. However, if the effective permeability of water increased more than the one of butanol, the separation factor for butanol would unfortunately decrease. The results of this analysis are presented in Figure 9 and confirmed what was expected. The relative permeability of water increased by only 3% for vertically aligned nanotubes, whereas it was lower than unity for the other orientations. The relative permeability of water decreased to 0.75 for horizontal nanotubes. For butanol, the relative permeability was about 0.88 for vertically aligned nanotubes and decreased to 0.76 for horizontal nanotubes. Based on the physical data of Table 3, it is clear that there would be no advantages using an MMM with nanotubes for the water-butanol separation. In addition, the butanol selectivity decreased with the addition of nanotubes. For horizontal nanotubes, the selectivity was identical to the one prevailing for the polymeric membrane, but the effective permeability was lower.

### 3.6. Comparison of the Relative Permeability with Existing Models

One of the objectives of this investigation was to assess the accuracy of existing empirical models on the prediction of relative permeability. Since the correlation proposed by Kang et al. [26] to estimate effective permeability, referred as the KJN model, is one that considers the orientation of the nanotubes and their diffusion coefficient in the axial direction, it was compared first to the data obtained in this investigation, considered to be the exact solutions. The KJN model was developed for MMMs with tubular fillers assuming a one-dimensional mass transport. In this section, the relative permeability *P_r_* was calculated using Equations (4)–(6) for an MMM with embedded nanotubes with an orientation varying from 0° to 90°, with a constant length-to-diameter aspect ratio of 10 and a filler volume fraction *φ* of 0.1. The results are presented in Figure 10 for three values of the permeability ratio (*P_d_*/*P_c_*). The results show that the KJN model predicted a nearly constant relative permeability over a very large range of orientation and its value only dropped at around 80°. In addition, the predicted values of the relative permeability were nearly independent of the permeability ratio. The insensitivity of the KJN model to the permeability ratio was due to the predominance of the first term of Equation (4) compared to the second term containing the permeability ratio. For a permeability ratio of 1000, the first term varied between 0.9 to 1.0, whereas the second term varied between 0.0001 to 0.001. Comparing the results of the numerical simulations for various values of the relative permeability, it is obvious that the KJN model cannot adequately predict the relative permeability of mixed-matrix membranes with nanotube fillers.

Since unreliable predictions were obtained with the KJN model, it was decided to perform a similar comparison with other empirical models that had the most predictive potential. The Maxwell model and the Hamilton-Crosser model were retained for this additional comparison. The Maxwell model was derived for MMMs containing spherical fillers with isotropic diffusivity, whereas the Hamilton-Crosser model was developed for MMMs with tubular fillers. The relative permeability *P_r_* was calculated by the two models and compared to the numerical results for vertically aligned tubular fillers with a constant length-to-diameter aspect ratio of 100 and three values of the permeability ratio (*P_d_*/*P_c_*) (10, 100, and 1000). The results of this study are presented in Figure 11. For a permeability ratio of 10 (Figure 11a), the diffusion barrier effect of the impermeable side wall of the nanotubes perpendicular to the main direction of the migrating species was more significant than the enhancement of the higher permeability in the axial direction. This diffusion barrier of the nanotube wall was only considered by the numerical method. As a result, the three models overpredicted the relative permeability for this lower permeability ratio. For the higher (*P_d_*/*P_c_*) ratios of 100 (Figure 11b) and 1000 (Figure 11c), the Maxwell and KJN models underestimated the results whereas the Hamilton-Crosser model overpredicted the relative permeability. Even though the Maxwell model underestimated the relative permeability, at a lower nanotube volume fraction and an intermediate permeability ratio (*P_d_/P_c_* = 100) it provided a relatively better prediction. Therefore, additional numerical simulations were performed to calculate the relative permeability as a function of the filler volume fraction with a permeability ratio of 100 for three length-to-diameter aspect ratios (*α* of 10, 20, and 50). The numerical results were compared to Figure 12 with the three prediction models. At a lower aspect ratio (*α* = 10), both the Maxwell and Hamilton-Crosser models predicted a higher relative permeability than the numerical simulations. On the other hand, for *α* of 20 and 50, the results of Figure 12b,c show that the numerically predicted relative permeability fell between the Hamilton-Crosser and Maxwell model predictions. However, the difference in *P_r_* between the present work and the estimations by the Maxwell model was relatively small for *P_d_*/*P_c_* = 100 and *α* = 10 and 20, but it became more pronounced for higher values of *P_d_*/*P_c_* and α. According to all the comparisons in this section, there is obviously a need to develop a new correlation that provides more accurate predictions of the relative permeability of MMMs embedding nanotube fillers.

## 4. Conclusions

The Fickian diffusion equation was solved numerically to estimate the effective permeability of MMMs with tubular fillers. The numerical model explicitly accounts for the effects of the filler permeability, volume fraction, aspect ratio, and orientation. Several general conclusions can be drawn from this study. First, the filler orientation has an enormous effect on the mixed-matrix membrane’s effective permeability and aligning the fillers vertically is very favourable. Second, in the vertically aligned applications, fillers with higher aspect ratios are more beneficial to MMMs and significantly increase the effective permeability. Third, fillers with a low dispersed-to-continuous phase permeability ratio (*P_d_/P_c_* < 10) should be avoided in the fabrication of mixed-matrix membranes due to the significant diffusion barrier effect of the impermeable side wall of the nanotubes perpendicular to the main direction of the migrating species, which is more than the enhancement of the higher permeability in the axial direction. On the other hand, fillers with a lower outside-to-inside diameter ratio are more advantageous. Finally, existing analytical models for the prediction of the relative permeability of MMMs with tubular fillers are not reliable. Therefore, there is a need to develop a new correlation that would provide more accurate predictions of the relative permeability of MMMs embedding nanotube fillers.

## Figures and Tables

**Figure 1 membranes-11-00058-f001:**
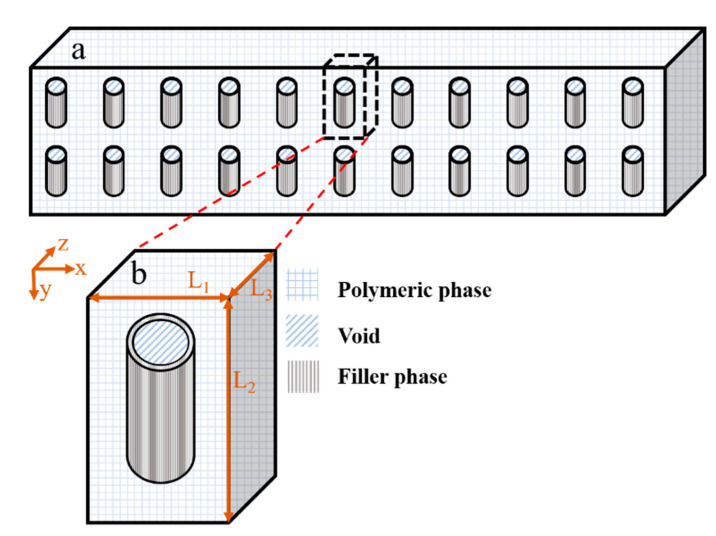
(**a**) A schematic diagram of the cross-section of homogenous dispersed nanotube particles in a mixed-matrix membrane and (**b**) a schematic diagram of a repetitive unit element including the polymer and filler phases as well as void volume inside the nanotube along with the parameter giving its dimensions.

**Figure 2 membranes-11-00058-f002:**
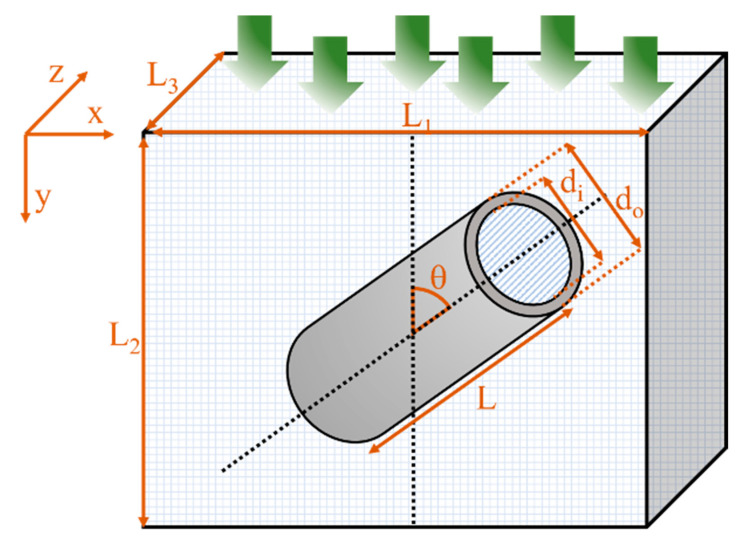
Schematic diagram of a repetitive unit element of a mixed-matrix membrane.

**Figure 3 membranes-11-00058-f003:**
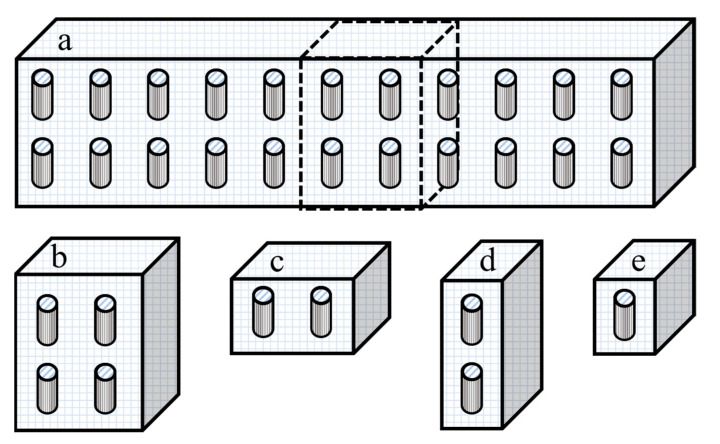
(**a**) A schematic diagram of a cross-section of the MMM showing a series of repeatable unit elements. A series of smaller membrane cross-sections: (**b**) four unit elements with vertical nanotubes (Case I), (**c**) two unit elements with vertical nanotubes distributed on the *x*-axis (Case II), (**d**) two unit elements with vertical nanotubes distributed on the *y*-axis (Case III), and (**e**) one unit element with a vertical nanotube at the center of the element (Case IV).

**Figure 4 membranes-11-00058-f004:**
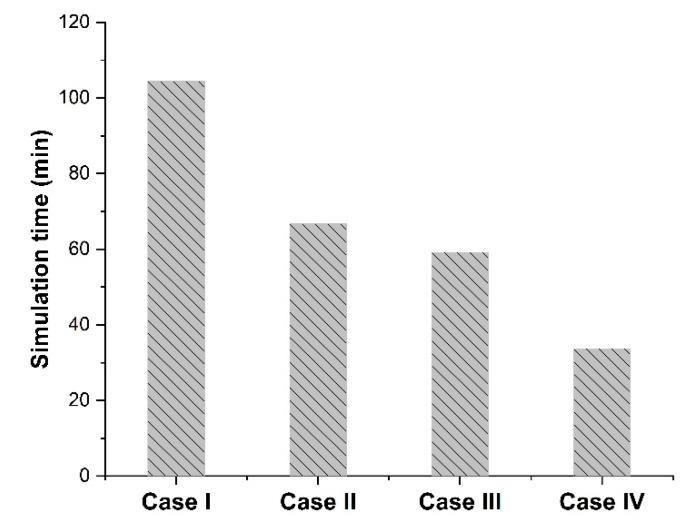
Computation times of the numerical simulations for the four different case studies using one core and 1 Gb RAM on the Frontenac cluster nodes available at the Centre for Advanced Computing (CAC) at Queens University (Canada).

**Figure 5 membranes-11-00058-f005:**
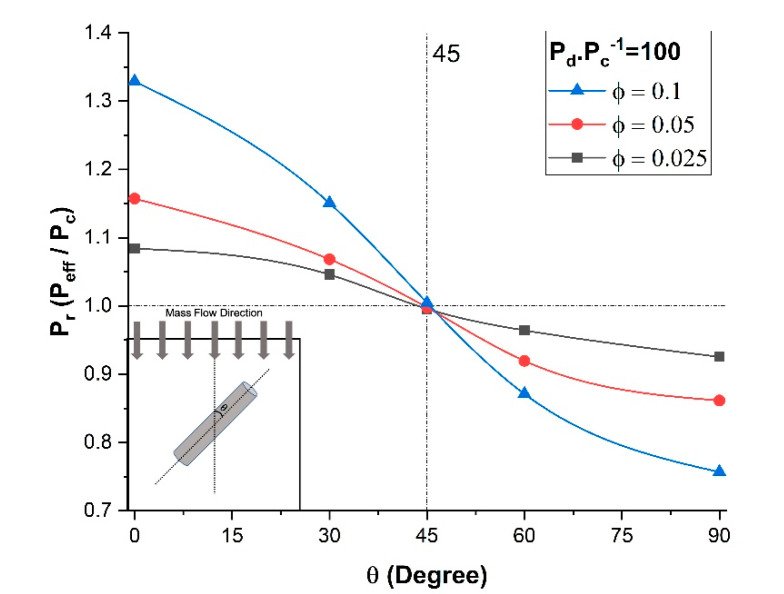
Relative permeability for different nanotube orientations and volume fractions at *P_d_*/*P_c_* = 100 and α = 10.

**Figure 6 membranes-11-00058-f006:**
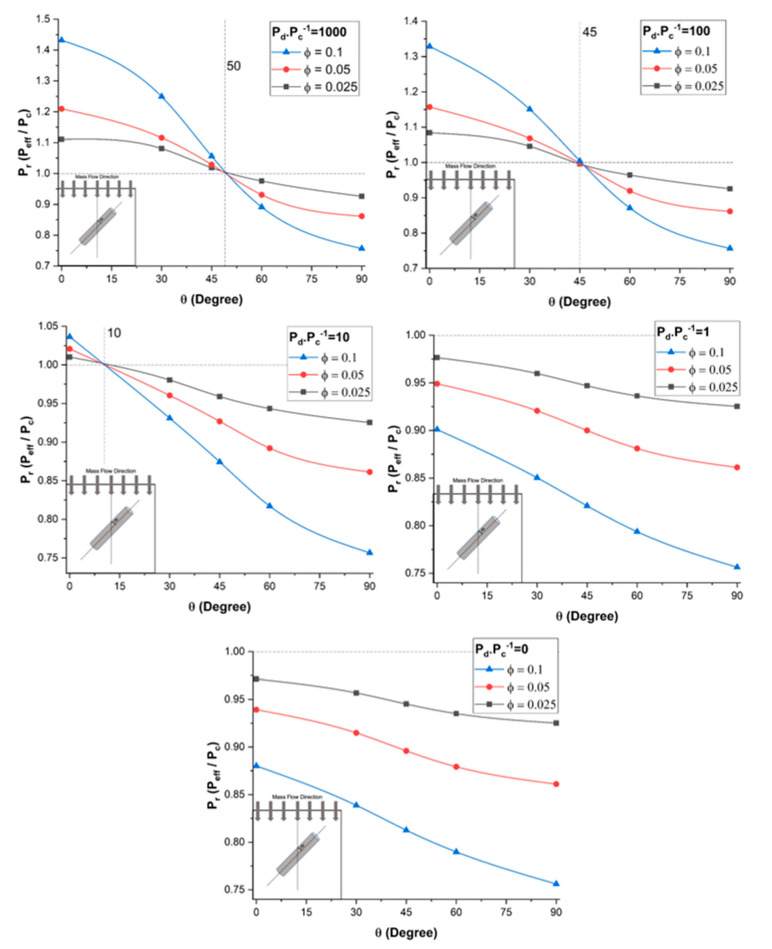
Plots of the relative permeability of mixed-matrix membranes incorporating nanotubes as a function of the nanotube orientation for five different *P_d_*/*P_c_* ratios, three filler volume fractions, and at a constant length-to-diameter aspect ratio *α* = 10.

**Figure 7 membranes-11-00058-f007:**
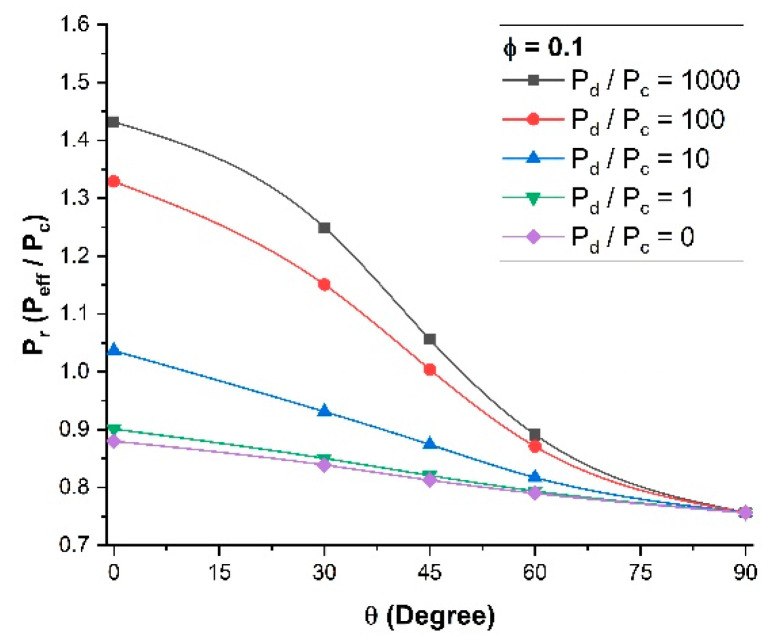
Relative permeability for different dispersed-to-continuous permeability ratios *P_d_*/*P_c_* at a constant volume fraction *φ* = 0.1 and a constant length-to-diameter aspect ratio *α* = 10.

**Figure 8 membranes-11-00058-f008:**
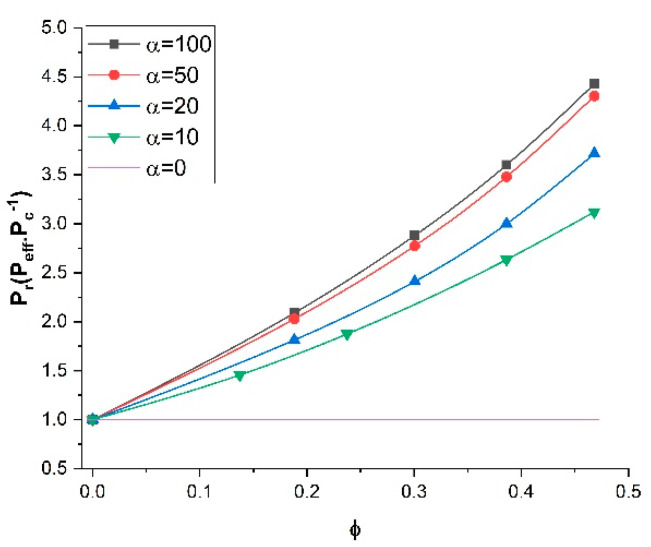
Relative permeability for nanotubes with different length-to-diameter aspect ratios at a permeability ratio *P_d_*/*P_c_* of 100 and an angle *θ* of 0°.

**Figure 9 membranes-11-00058-f009:**
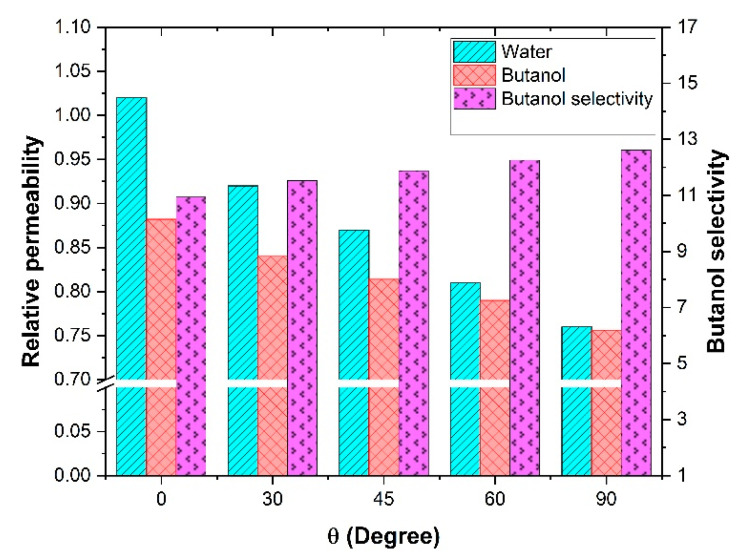
Water and butanol relative permeability and butanol selectivity at different orientations of the nanotubes for a length-to-diameter aspect ratio *α* of 10.

**Figure 10 membranes-11-00058-f010:**
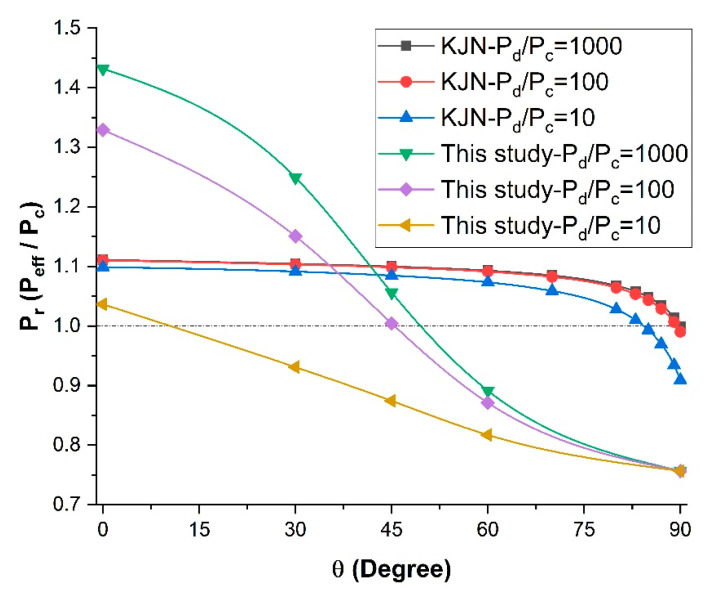
Comparison of the relative permeability predicted by the numerical simulations with the KJN model as a function of the nanotube orientation for three permeability ratios at *φ* = 0.1 and *α* = 10.

**Figure 11 membranes-11-00058-f011:**
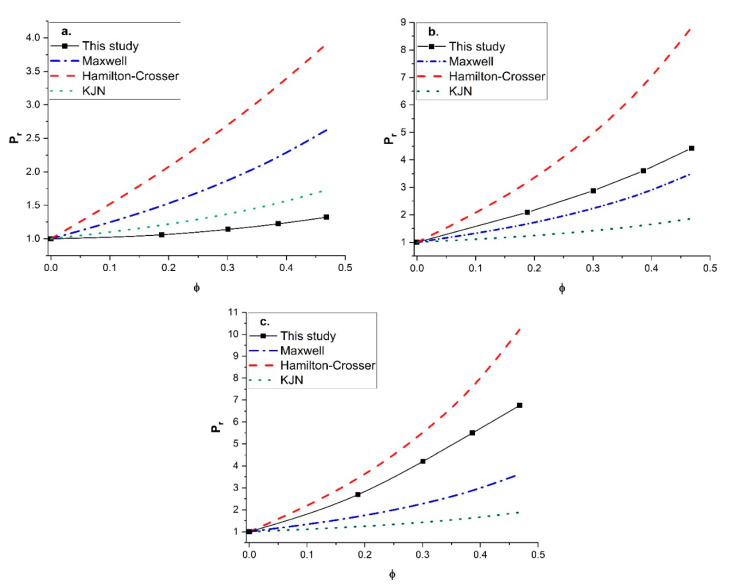
Comparison of the predicted relative permeability obtained in this study with the Maxwell, Hamilton-Crosser and KJN models as a function of the nanotube volume fraction for *α* = 100: (**a**) (*P_d_*/*P_c_*) = 10, (**b**) (*P_d_*/*P_c_*) = 100, and (**c**) (*P_d_*/*P_c_*) = 1000.

**Figure 12 membranes-11-00058-f012:**
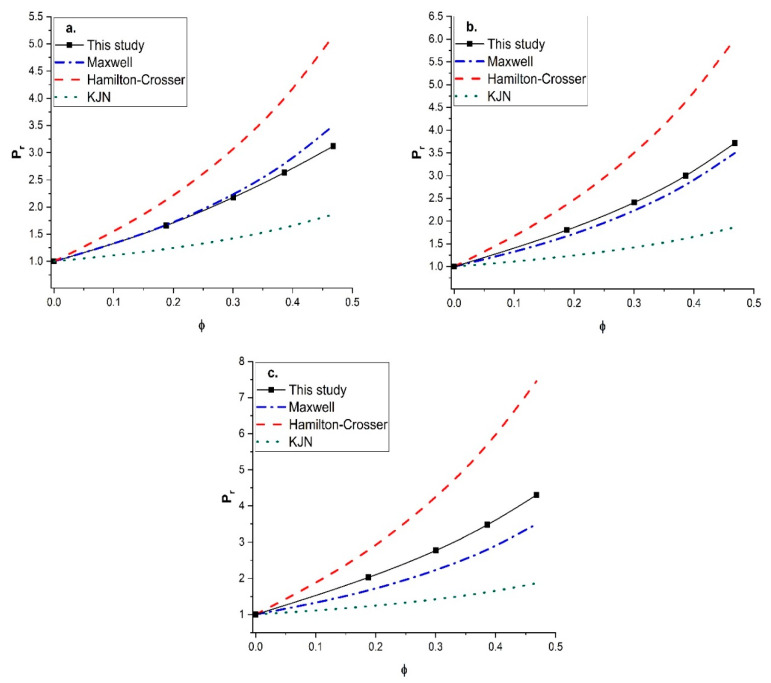
Comparison of the predicted relative permeability obtained in this study with the Maxwell, Hamilton—Crosser, and KJN models as a function of the nanotube volume fraction for (*P_d_*/*P_c_*) = 100: (**a**) *α* = 10, (**b**) *α* = 20, and (**c**) *α* = 50.

**Table 1 membranes-11-00058-t001:** Initial and boundary conditions used in all simulations.

**Initial Condition**	
t=0	${C_{\left(x,y,z\right)}|}_{t=0}=C_{\left(x,0,z\right)}-\frac{C_{\left(x,0,z\right)}-C_{\left(x,L_{2},z\right)}}{N_{y}-1}$
**Boundary Conditions**	
y=0	$C_{\left(x,0,z\right)}=C_{f}.S_{\left(x,0,z\right)}$
$y=L_{2}$	$C_{\left(x,L_{2},z\right)}=0$
$x=0\;\&\;x=L_{1}$	${\frac{\partial C}{\partial x}|}_{x=0}={\frac{\partial C}{\partial x}|}_{x=L_{1}}=0$
$z=0\;\&\;z=L_{3}$	${\frac{\partial C}{\partial z}|}_{z=0}={\frac{\partial C}{\partial z}|}_{z=L_{3}}=0$

**Table 2 membranes-11-00058-t002:** Dispersed and continuous phase specifications, which are representative of single-wall carbon nanotubes.

Case	ϕ	Number of Unit Elements	Simulation Dimension L_1_ × L_2_ × L_3_ (nm)	Aspect Ratio α	Outside Diameter *d_o_* (nm)	Inside Diameter *d_i_* (nm)	Dispersed-to-Continuous Permeability Ratio (*P_d_*/*P_c_*)
I	0.1	4	10.6×44×5.3	10	2.0	0.8	100
	0.4		5.4×44×2.7				
II	0.1	2	10.6×22×5.3				
	0.4		5.4×22×2.7				
III	0.1	2	5.3×44×5.3				
	0.4		2.7×44×2.7				
IV	0.1	1	5.3×22×5.3				
	0.4		2.7×22×2.7				

**Table 3 membranes-11-00058-t003:** Summary of simulation details for the continuous and dispersed phases.

Temperature (K)	310	
Filler Volume Fraction (*φ*)	0.1	
Filler aspect ratio (*α*)	10	
Filler outer diameter (nm)	2.0	
Filler inner diameter(nm)	0.8	
	Water	Butanol
Feed concentration (kg·m^−3^)	992.5	5.0
D_c_ (m^2^·s^−1^)	$5.27\times 10^{-10\;}$ [1]	$7.29\times 10^{-10}\;$ [1]
D_d_ (m^2^·s^−1^)	$1.6\times 10^{-7}$	$7.94\times 10^{-8}$
S_c_ (g·m^−3^/g·m^−3^)	$1.6\times 10^{-3\;}$ [4]	$1.46\times 10^{-2}\;$ [4]
S_d_ (g·m^−3^/g·m^−3^)	$4.44\times 10^{-5}$	$1.38\times 10^{-5}$
P_c_ (m^2^·s^−1^)	$8.43\times 10^{-13}$	$1.06\times 10^{-11}$
P_d_ (m^2^·s^−1^)	$7.10\times 10^{-12}$	$1.10\times 10^{-12}$

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
