# Peer review of "Modelling the Molecular Permeation through Mixed-Matrix Membranes Incorporating Tubular Fillers"

_membranes, 2021, doi:10.3390/membranes11010058_

Round 1
Reviewer 1 Report
The paper of Zamani et al. discusses the permeation in mixed matrix membranes, prepared by tubular fillers solving the Fick diffusion equation.
This is a quite interesting academic exercise, but the authors should state more clearly the importance of this work and they should combine it better with some experimental value, even from literature.
As a minor remarks, I suggest to check a bit better the definition, since this reviewer does not understand what is the mean of “permeation flux at the feed side” (Page 6, line 216).
I also suggest not use alpha as “length-to-diameter ratio” since it is commonly associated to selectivity in the field of membrane and this leads to misleading.
Page 9, Line 273 this is a quite obvious statement since it is the validity of P=D*S, and in many cases the authors discuss common knowledge of the sector in details, which dilute the interesting sections of the paper, such as the break-even angle. Thus, I encourage the authors to delete the well-known discussion and to focus more on the real results of their work.
Reviewer 2 Report
In “Modeling the molecular permeation through mixed matrix membranes incorporating tubular fillers” the Fick diffusion equation is solved in 3D to study the effect of orientation, aspect/ratio and disperse-to-continuous permeability over the performance of Polymer/nanotubes MMMs. The work is interesting and well explained. I recommend accepting it after minor revisions.
I have the following comments to the authors:
1) When discussing the selectivity permeability trade-off, the Robeson’s upper bound paper of 2008 should be cited.
2) “It has been reported that MMMs, which are made by embedding a proper organic or inorganic filler in the polymer matrix, can combine the advantages of the higher selectivity of the filler particles and the ease of processing of polymers” here the leaders in the field should be cited (for example: Koros, Caro, Kapteijn, Nair….).
3) References 10-11-12 should be replaced or complemented by recent reviews in the field, for instance, for reference 12, I suggest: Chem. Rev. 2020, 120, 16, 8267–8302.
4) “However, several additional parameters such as particle pore size and distribution, permeability of species in the rigidified or void layer, filler pore blockage ratio as well as the thickness of the non-ideal phase should be taken into account. These parameters are most sensitive to operating conditions and, since there are no reliable methods yet to measure these parameters, their estimation and determination remain a significant challenge”: the authors may wish to cite some of the atomistic simulation studies that have captured many of those parameters with high detail recently, for example: Chem. Mater. 2020, 32, 3, 1288–1296.
5) Line 163: “In this investigation, it was assumed that the particle-filler interface morphology was ideal”. It would be great if the authors give more detail on what they mean for ideal, for example: no interface voids, no polymer penetration or blockage of the pore entry…
6) It would help to put Figure 2 earlier in the text, as it is important for the reader to be able to quickly identify that the “y” axis is the axis parallel to the orientation of the nanotube (or alternatively, add the x y z axes to Figure 1b).
7) Line 187: I don’t understand what the authors mean by “mesh independency”. Is it to avoid non-physical self-interactions? A more specific description would help.
8) Eq (12): in the definition of wi, it should read “0.5, i=1 or Nx” instead of “0.5, i=1 or Ni”
9) L2 is only defined at line 213, while it has been introduced much earlier in the equations
10) How valid do authors think that assuming Knudsen diffusion inside the nanotube is? It would be great if they could comment this assumption made in lines 220/221.
11) Did the authors consider investigating the distance between nanotubes as a variable?
12) Line 240: Is Pd/Pc = 100 a representative value? Do authors have a concrete physical system in mind?
13) In Table 2, a reminder that do is the outer diameter of the nanotube and di the inner diameter of the nanotube would be helpful for the reader, since they are defined long before. Also how were those values chosen? Are they representative for a particular physicochemical system?
14) Line 272: authors could also mention that this is also the case for nanosheets as fillers, and not only for nanotubes. MMMs with oriented nanosheets have been shown to improve adsorption performance with respect to non-oriented nanosheet-based MMMs.
15) “In this investigation, it was assumed that, given the concentrations within the membrane are relatively low, there are no interactions effects between the two migrating species.” How valid is this assumption inside the nanotubes? It seems to me that local concentration will be high due to confinement effects.
16) When the authors compare their model results with other models results (KJN, Maxwell, Hamilton-Crosser) they should report experimental values in order to decide which model gives the most physically sound description of reality. Without these experimental results, it is not obvious to me that their model is better, surely it is more sophisticated, but does it give predictions that are closer to reality?
17) “According to the all comparisons in this section, there is obviously a need to develop a new correlation that will provide more accurate predictions of the relative permeability of MMMs embedding nanotube fillers.” I don’t understand this statement, are the authors saying that their model is still not good enough? If this is the case, could they explain what could be improved?
Reviewer 3 Report
The work is based on modeling in MMMs for separation application by extensive studying various aspects of filler orientations, loadings, and parameter in polymer matrix, against relative permeability. Separation properties predictions of binary mixtures and comparison with existing models were also studied. Following recommendations are suggested.
- The manuscript lacks quality in terms of similarity/plagiarism with previously published sources. Grammatical errors and typo mistakes are observed. Revise the manuscript to address the issues.
- The abstract needs to be revised and more focused on current work and results obtained.
- AARE% must be tabulated for each comparative permeability throughout the figures in the manuscript.
Reviewer 4 Report
Manuscript No.: Membranes-1018230
Title: Modeling the molecular permeation through mixed-matrix membranes incorporating tubular fillers
The current work developed a steady-state concentration profile, which was calculated by solving numerically the Fick diffusion equation, to simulating the effective permeability of nanotube-incorporated polymeric mixed matrix membrane. However, this modeling didn’t lead to performance enhancement in terms of permeability and selectivity for water-butanol separation when PDMS/nanotubes mixed-matrix membranes were used. In addition, even the results were then compared with Maxwell, Hamilton-Cross and Kang-Jones-Nair models, those predicted results could not be validated by experimental data obtained from previous work. Therefore, it is a pity, this modeling cannot be recognized as a useful tool for understanding and designing MMM with tubular fillers, if the following issued could be address by the authors.
- In abstract, line 27, the authors mentioned that “the permeability increased with the permeability ratio, ……” What’s permeability ratio mean?
- what’s the Why the empirical parameter q was taken as 1? The authors should clearly define the hypotheses adopted for the modeling derivation in this paper.
- The values of f was too low to simulate the performance of MMMs. So far, the high loading fillers of MMMs have been developed.
- The experimental data should be used to fit the model developed here to evaluate its accuracy.
Round 2
Reviewer 1 Report
This reviewer still thinks that this is a quite interesting academic exercise, but the authors should state more clearly the importance of this work.
Reviewer 4 Report
The authors have adequately addressed my previous comments and suggestions. In addition, the revision is satisfactory and the changes are acceptable.
